# Near-Infrared Fluorescence Imaging of Renal Cell Carcinoma with ASP5354 in a Mouse Model for Intraoperative Guidance

**DOI:** 10.3390/ijms23137228

**Published:** 2022-06-29

**Authors:** Katsunori Teranishi

**Affiliations:** Graduate School of Bioresources, Mie University, 1577 Kurimamachiya, Tsu 514-8507, Mie, Japan; teranisi@bio.mie-u.ac.jp; Tel.: +81-59-231-9615

**Keywords:** ASP5354, diagnosis, fluorescence imaging, real-time imaging, renal cell carcinoma

## Abstract

Renal cell carcinoma is a prevalent disease associated with high morbidity and mortality rates. Partial nephrectomy is a first-line surgical option because it allows the preservation of renal function. Clear differentiation between normal and cancerous tissues is critical for increasing the negative margin rates. This study investigated the capability of the near-infrared (NIR) fluorescent imaging agent ASP5354 for in vivo fluorescence imaging of renal cell carcinoma. ASP5354 at a single dose of 12 nmol (0.037 mg)/kg body weight was intravenously administered to healthy and orthotopic renal cell carcinoma mice under anesthesia. NIR images of the abdominal cavity were obtained using a near-infrared fluorescence (NIRF) camera system. In addition, the cancerous kidneys were harvested, and the NIRF in their sections was measured using an NIRF microscope. Normal renal tissue emitted strong NIRF but the cancer tissue did not. The difference in NIRF intensity between the normal and cancer tissues clearly presented the boundary between the normal and cancer tissues in macro and micro NIRF imaging. ASP5354 can distinguish cancer tissue from normal tissue using NIRF. Thus, ASP5354 is a promising agent for renal cell carcinoma tissue imaging in partial nephrectomy for renal cell carcinoma patients.

## 1. Introduction

Renal cell carcinoma is a prevalent disease associated with high morbidity and mortality rates. In 2020, a total of 431,288 incident cases of renal cell carcinoma were diagnosed worldwide [1]. In the same year, 179,368 renal cell carcinoma–related deaths were reported [1]. Diagnostic advances have enabled the early detection of renal cell carcinoma. Malignant renal tumors are mainly treated with either partial or radical nephrectomy. Partial nephrectomy is the first-line surgical option and is preferred over radical nephrectomy for small renal masses as it allows the possible preservation of renal function [2,3]. An effective partial nephrectomy requires a high negative margin rate. Thus, advanced diagnostic modalities, including ultrasonography, pathologic assessments, and optical imaging such as fluorescence imaging are crucial for this surgery [4,5,6].

Intraoperative fluorescence image–guided surgery in urology has been actively studied [7,8]. Intraoperative fluorescence imaging requires light penetration to and from deep tissue, low autofluorescence, and low light absorption and scattering by tissue, all of which can be achieved using near-infrared (NIR) light at 700−900 nm [9]. Indocyanine green (ICG), which fluoresces in the NIR region and is approved by the Food and Drug Administration, is the most widely used imaging agent for lymph node identification, blood vessel imaging, and tumor imaging [10]. However, the NIR fluorescence (NIRF) intensity of ICG is affected by protein and pH [11].

ICG accumulates in the liver following intravenous administration and can thus be useful in the identification of liver cancer [12,13]. Further, ICG transfers to the kidneys in negligible amounts. Importantly, the fluorescence intensity of ICG is lower in malignant renal tissue than in normal renal tissue, and thus, it is useful in imaging renal cell carcinoma for partial nephrectomy [6,14,15,16,17,18,19]. However, ICG is associated with adverse reactions [20]. Chu et al. reported anaphylactic shock during robotic partial nephrectomy using intravenous ICG administration [21]. Moreover, the efficacy of ICG imaging–guided partial nephrectomy differs among studies [22,23].

Folate receptor expression differs between malignant and normal renal tissues. In the kidney, folate receptor expression is higher in normal tissues than in tumor tissues. Guzzo et al. reported that the fluorescence imaging compound EC17, which comprises folate and fluorescein, identified renal cell carcinoma during surgery in four patients [24]. Shum et al. also demonstrated the utility of the NIR fluorescent compound OTL38, in which folate is bound to the NIR fluorescent heptamethine cyanine compound, in robot-assisted laparoscopic partial nephrectomy [25]. Given the low binding capability of EC17 and OTL38 to renal tumor tissue in the kidney, the fluorescence intensity is lower in malignant tissue than that in normal tissue. However, these probes have no specific features for renal accumulation and clearance following their intravenous administration, and thus, high-dose administrations are needed.

A new NIRF imaging probe, CD-NIR-1, and its more chemically stable analog ASP5354 (Figure 1), have recently been reported for intraoperative ureteral identification and diagnosis [26,27]. CD-NIR-1 and ASP5354 emit fluorescence at *λ*_max_ 801 nm and 815 nm, respectively, in phosphate-buffered saline (PBS, pH 7.4), allowing deep tissue penetration of light, low autofluorescence from tissue, and low light absorption and scattering by tissues, making them suitable for in vivo optical imaging of living tissue. Moreover, these probes are transferred post-intravenous administration into urine through specific and ultra-rapid renal clearance with no chemical modification, as shown in rat, minipig, and macaque models. Therefore, these probes are suitable for renal imaging after intravenous administration. Recently, the safety/tolerability and pharmacokinetics of ASP5354 have been assessed in its first human phase 1 trial [28]. This study aimed to investigate the usefulness of ASP5354 as an imaging agent for NIRF differentiation between renal cell carcinoma and normal tissues in a mouse model.

## 2. Materials and Methods

### 2.1. Animals

This study complied with the regulations for animal experiments of the Mie University Animal Ethics Committee and the Institutional Animal Care and Use Committee at ITECHLAB Co., Ltd. (Gifu, Japan). BALB/cCrSlc mice (female; age, 7 weeks; mean weight, 20 g) were purchased from Japan SLC, Inc. (Shizuoka, Japan). All mice were housed under specific pathogen-free conditions prior to the experiments. The experiments were performed after all mice were anesthetized using subcutaneous injections of ketamine (75 mg/kg) and medetomidine (1 mg/kg).

### 2.2. Preparation of Orthotopic Renal Cell Carcinoma Mouse Model

RenCa cells were cultured in RPMI-1640 containing 10% fetal bovine serum (FBS) and an antibiotic solution (penicillin and streptomycin) in an incubator at 37 °C and 5% CO_2_. The resulting cells were washed with RPMI-1640, and the cell suspensions (1.3 × 10^7^ cells/mL) were prepared. After anesthetization, the mice were placed in the lateral position for implantation of RenCa cells, and the skin was opened using a left flank incision. The left kidney was exposed and RenCa cells (1.3 × 10^7^ cells/mL, 0.04 mL) were injected into the subcapsular space of the kidney using a 27G needle. The kidney was returned to the peritoneal space, and the incision was sutured. NIRF imaging was performed after 9 and 14 days.

### 2.3. Materials

ASP5354 (C_135_H_197_N_4_O_73_Cl, molecular weight: 3079), formerly termed TK-1, was prepared as previously described [29]. ICG was purchased from MP Biomedicals, LLC (Solon, OH). RenCa cells (murine renal carcinoma cell line) were purchased from CLS Cell Lines Service GmbH (Eppelheim, Germany). RPMI-1640 was purchased from FUJIFILM Wako Chemicals Co., Ltd. (Osaka, Japan), penicillin–streptomycin solution (5000 U/mL) from Thermo Fisher Scientific K.K. (Tokyo, Japan), FBS from MP Bio Japan (Tokyo, Japan), ketamine from Daiichi Sankyo Propharma Co., Ltd. (Tokyo, Japan), and medetomidine from Kyoritsu Seiyaku Co., Ltd. (Tokyo, Japan).

### 2.4. Instruments

Real-time NIRF imaging was performed using a photodynamic Eye-neo (PDE)-neo camera system (Hamamatsu Photonics K.K., Shizuoka, Japan), optimized for ICG and equipped with a 760-nm light-emitting diode for excitation and a charge-coupled device for detection. In addition, an optical high-pass filter for NIRF detection was placed in front of the charge-coupled device detector. Video images were recorded using a personal computer. Measurement conditions were as follows: brightness, −3.0; contrast, 5; irradiation intensity, 7.5. NIRF intensity was analyzed using the region of interest (ROI) analysis program (Hamamatsu Photonics K.K.). The intensities were shown as means in the ROI. Microscopic observation of tissue sections and RenCa cells was performed using an Axiovert 200 microscope (Carl Zeiss Co., Ltd., Oberkochen, Germany) equipped with an object lens Plan-Apochromat 20×/0.75 (Carl Zeiss Co., Ltd.) and a monochrome camera (Axio CamMRm; Carl Zeiss Co., Ltd.) at 20 °C. NIRF was measured using the filter set 41037 Li-Cor for IR Dye 800 (excitation bandpass: 720–760 nm, emission long-pass: >780 nm; Chroma Technology, Bellows Falls, VT, USA), and images were obtained using AxioVision 4.8 software (Carl Zeiss Co., Ltd.). An ECLIPSE E600 microscope (Nikon Corporation, Tokyo, Japan) equipped with a camera (E8400, Nikon Corporation) was used to observe the sections stained with hematoxylin and eosin (H&E).

### 2.5. ASP5354 NIRF Imaging

After anesthetizing healthy and renal cell carcinoma mice, ASP5354/saline (2.4 μM) was intravenously injected at 12 nmol (0.037 mg)/kg body weight in the tail. After laparotomy, NIRF imaging of the internal organs was performed in the dark using a PDE-neo camera system at 10 min (*n* = 3 for healthy mice, *n* = 5 for cancer mice) and 6 h (*n* = 3 for healthy mice, *n* = 3 for cancer mice) following ASP5354 administration. After the in vivo NIRF imaging of cancer mice, the mice were sacrificed, the cancerous kidney was excised, and NIRF imaging of the cut surfaces of the kidney was performed in the dark using the PDE-neo camera system.

### 2.6. Histopathological Analysis and NIRF Imaging of Cancerous Kidney Sections

After in vivo NIRF imaging following intravenous administration of ASP5354 (12 nmol/kg body weight), the mice were sacrificed and the cancerous kidney was excised. The tissues were then frozen and sectioned. Next, NIRF imaging of the sections was performed using an Axiovert 200 microscope in the dark and displayed in white. Additionally, another frozen section adjacent to the section used for NIRF observation was stained with H&E and observed under a microscope.

### 2.7. NIRF Imaging of Cellular Uptake of ASP5354

A suspension of RenCa cells (1 × 10^5^ cells/mL) was incubated in 0.5 mL of a saline solution of 2.4- or 24-μM ASP5354 at 37 °C for 10, 30, or 60 min. Next, the incubated solution was centrifuged (500 rpm, 20 °C, 5 min), and the obtained cells were washed five times with 0.5 mL PBS. The centrifuged cells were then suspended in 0.1 mL PBS (pH 7.4), and the cell suspension was observed under an Axiovert 200 microscope equipped with an object lens Plan-Apochromat 20×/0.75 and a monochrome camera (Axio CamMRm) at 20 °C. Next, NIRF was measured using the filter set 41037 Li-Cor for IR Dye 800 (excitation bandpass: 720–760 nm, emission long-pass: >780 nm). NIRF was measured for 60 s in a 1 × 1 binning mode, and images were obtained using AxioVision 4.8 software.

### 2.8. ICG NIRF Imaging

After anesthetizing renal cell carcinoma mice, ICG/saline (24 μM) was intravenously injected at 120 nmol/kg body weight in the tail. After laparotomy, NIRF imaging of the internal organs was performed in the dark using a PDE-neo camera system at 10 min (*n* = 3) following ICG administration.

### 2.9. Statistical Analysis

Data are reported as the mean ± standard deviation. Statistically significant differences were determined using Student’s *t*-test.

## 3. Results

### 3.1. In Vivo NIRF Imaging of ASP5354 Clearance

At 10 min after ASP5354 injection at 12 nmol/kg body weight (this dose was set based on previous experiments [26]) into the tail of healthy BALB/cCrSlc mice, NIRF of ASP5354 was selectively emitted from the kidneys and bladder more strongly than from other organs and tissues (Figure 2). This result indicated that ASP5354 was selectively and rapidly transferred into the kidneys and subsequently into the urine in the BALB/cCrSlc mice. Moreover, an ASP5354 single dose of 12 nmol/kg body weight was demonstrated to be suitable for clear NIRF imaging of the kidneys using a PDE-neo camera system.

### 3.2. In Vivo and Ex Vivo NIRF Imaging of Renal Cell Carcinoma with ASP5354

Based on the finding that ASP5354 was transferred into the kidneys selectively and rapidly following intravenous administration, this study assessed in vivo NIRF imaging of cancerous kidneys. At 9 days and 14 days following RenCa cell implantation into the subcapsular space of the healthy kidney, post-laparotomy NIRF imaging of cancerous and normal kidneys was performed at 10 min following the ASP5354 intravenous administration of a dose of 12 nmol/kg body weight. NIRF was uniformly emitted from the entire normal kidneys (Figure 2b); however, NIRF imaging showed no or low NIRF intensity in cancer tissues. Larger cancer tissues that had non-fluorescence was easily distinguished from the NIR fluorescent normal tissue (Figure 3a–f). NIRF imaging revealed that cancer tissue less than 1 mm in diameter could be detected as lowly fluorescent tissues (Figure 3g). Ex vivo NIRF imaging of the section surfaces of the cancerous kidneys also clearly demonstrated the NIRF difference between NIR fluorescent normal tissues and NIR non-fluorescent cancer tissues (Figure 3b,d). ROI analysis of NIRF intensity showed the clear difference between the normal and cancerous tissues (Figure 3h). The NIRF intensities in ROI in the normal and cancerous areas, as shown in Figure 3h, were 82 ± 20 and 6.4 ± 11 (mean ± standard deviation), respectively, and the data were considered to be statistically significant (*p* < 0.001). At 6 h after ASP5354 administration, no NIRF was emitted from either healthy or cancerous kidneys (Figure 4), indicating that ASP5354 did not accumulate in the cancer tissue after ASP5354 administration.

ASP5354 fluorescence in the renal tissue at 10 min following ASP5354 administration was confirmed by microscopic NIRF observation. No NIRF was found in the cancer tissue and the normal tissue clearly emitted NIRF (Figure 5). The boundary between the normal and cancerous areas was clearly demonstrated by the difference in NIRF intensity.

### 3.3. NIRF Imaging of Cellar Uptake of ASP5354

The cellular uptake of ASP5354 to RenCa cells was assessed by incubating the cells in 24 µM ASP5354 solution, followed by NIRF imaging using a NIRF microscope (Figure 6). No cellular NIRF signal was observed after 10 min incubation, but 30 and 60 min incubation exhibited cellular NIRF signals. No NIRF signal from cells was obtained after 30 min incubation in 2.4-μM ASP5354 (not shown). These experiments revealed that 24 µM ASP5354 could be taken up by RenCa cells within 30 min, with a microscopically detectable signal.

### 3.4. In Vivo NIRF Imaging of Renal Cell Carcinoma with ICG

When ICG was intravenously administration at 12 nmol/kg body weight, significant NIRF of ICG was not emitted in normal and cancerous renal tissues because the ICG was immediately accumulated in the liver and did not transfer to the kidneys. Subsequently, a single dose of ICG at 120 nmol/kg body weight offered a suitable NIRF signal in the kidneys (Figure 7a–d). Under this condition for the NIRF imaging of cancer tissues, NIRF contrast ratios between cancer and normal tissues were too low to clearly differentiate between cancer and normal tissues. Visible cancer tissues about 1 mm in diameter were not detected with ICG (Figure 7a).

## 4. Discussion

Intraoperative NIRF imaging guidance is a powerful tool for the treatment of renal cell carcinoma. The utility of ICG has been studied in partial nephrectomy [6]. The negligible transfer of ICG into the kidneys following intravenous administration can reduce its efficacy and reliability in renal cell carcinoma imaging [22,23]. Moreover, although the use of NIR fluorescent folate-bound OTL38 has been studied based on low folate receptor expression in renal cell carcinoma tissue, OTL38 cannot selectively transfer into the kidneys following intravenous administration [25].

The present study evaluated the capability of ASP5354 for real-time NIRF imaging of renal cell carcinoma using a mouse model orthotopically implanted with a RenCa murine renal carcinoma cell line. The results showed the selective and ultra-rapid renal transfer of ASP5354 following its intravenous administration in the BALB/cCrSlc mice used for the renal cell cancer model. This is in agreement with previous results in other animal studies using rats and minipigs [27]. Intravenous injection of ASP5354 at a dose of 12 nmol/kg body weight was indicated to be an effective administration method for NIRF imaging of the kidney using the clinically available imaging device PDE-neo. After ASP5354 administration, the NIRF of ASP5354 was immediately emitted from normal renal tissue; however, cancer tissue did not emit NIRF.

This difference in ASP5354 NIRF between the normal and cancer tissues clearly presented the boundary between the normal and cancer tissues. Furthermore, cancer tissue less than 1 mm in diameter was identified. In addition, there was no long-term NIRF emission in cancer tissues. These results suggest that ASP5354 can be used for the intraoperative identification of renal cell carcinoma at any time during partial nephrectomy immediately after the administration of the imaging agent. The safety and efficacy of the intravenous injection of ASP5354 show that this imaging method can be applied in different clinical settings.

Several NIRF imaging agents for renal cell carcinoma imaging, aside from ICG and OTL38, have been studied. The NIR fluorescent heptamethine cyanine compound IR-783 [30] and the monoclonal antibody Girentuximab-IRDye800CW [31] have been reported to exhibit human renal tumor cell–specific uptake and retention in in vitro cell experiments and in vivo renal cell carcinoma mouse experiments. An in situ self-assembled NIR peptide probe successfully demonstrated NIRF imaging of human renal cell carcinoma in an orthotopic renal cell carcinoma xenograft mouse model and ex vivo human cancer kidney experiment [32]. ICG conjugated with a 2.1 kDa PEG was reported as a renal-tubule-secreted NIRF imaging probe for hyperfluorescence imaging of renal cell carcinoma in a study using an orthotopic renal cell carcinoma mouse model [33]. However, except for ICG, no other clinically available NIRF imaging agents have been approved for renal cell carcinoma imaging in partial nephrectomy. In this study, use of ICG showed that the NIRF contrast ratios between cancer and normal tissues were too low to clearly differentiate between cancer and normal tissues. Therefore, more efforts are needed to develop suitable imaging agents for clinical use. This study confirmed that ASP5354 has potential clinical utility in renal cell carcinoma imaging. In a previous study, daily intravenous administration of ASP5354 for 4 weeks in cynomolgus monkeys found no toxic changes at doses of up to 300 mg/kg per day [27]. Therefore, ASP5354 may be a safe imaging agent for renal cell carcinoma.

ASP5354 is a low-molecular-weight compound of 3 kDa that has selective and ultra-rapid renal clearance after intravenous administration. ASP5354 circulates in the bloodstream, is effectively filtered in the kidneys, and is subsequently concentrated in the urine, as demonstrated previously [27]. The stronger NIRF from the kidneys and bladder than from other organs and tissues is caused by the concentration of ASP5354. In this study, RenCa cells took up ASP5354 in 24 μM ASP5354 solution within 30 min, but this was not significant for 2.4 μM ASP5354. Thus, intravenous administration of 0.1 mL of 2.4 μM ASP5354 solution in mice at 20 g body weight could not cause the uptake of ASP5354 in RenCa cells in the kidney because of the low concentration of ASP5354 in the mouse body.

The ultra-rapid renal clearance of ASP5354 prevented its accumulation in the RenCa carcinoma tissue, resulting in the suppression of ASP5354 uptake by RenCa cells. In addition, because RenCa carcinoma tissue has no urine production function, ASP5354 was not concentrated in the RenCa carcinoma tissue, and the tissue could emit no NIRF from ASP5354. Therefore, the cancer tissues were clearly differentiated from normal tissues using NIRF. In addition to cancer identification, this unique imaging mechanism may offer the advantage of detecting renal dysfunction in NIRF-guided surgery. The optimal dose of ASP5354 should be determined for each patient. Given that NIRF renal cell carcinoma imaging with ASP5354 can be carried out immediately after the intravenous administration of ASP5354, ASP5354 should be initially administered at a single standard dose, but re-administration is possible if imaging needs to be optimized.

This study has some limitations that need to be acknowledged. The study demonstrated the efficacy of ASP5354 in renal cell carcinoma imaging in an orthotopic renal cell carcinoma mouse model, and thus, the efficacy and safety of this technique in humans still need to be established. In this study, NIRF imaging experiments were performed using a clinically available NIRF device (PDE-neo camera system) for laparotomy. The mice experiments demonstrated that the appropriate ASP5354 dose was 12 nmol/kg body weight. However, it should be noted that the optimal dose of ASP5354 depends on the device employed, adjustment of the sensitivity and distance between objects, excitation source, and camera. Therefore, in the actual clinical settings of laparotomy, laparoscopy, and robotic-assisted surgery, the optimal dose should be determined according to the clinical situation. Confirmation of the validity of ASP5354 for renal cell carcinoma identification in clinical trials can have important implications for partial nephrectomy.

## 5. Conclusions

ASP5354 allowed the visualization of renal cell carcinoma tissues immediately after a single intravenous dose, using the NIRF camera system, in renal cell carcinoma mouse model. Normal renal tissues emitted strong NIRF but cancer tissues did not. The difference in NIRF intensity between normal and cancer tissues clearly presented the boundary between the normal and cancer tissues in macro and micro NIRF imaging. Thus, ASP5354 is an effective imaging agent for renal cell carcinoma based on NIRF differentiation between normal and cancer tissues. ASP5354 is a promising agent for renal cell carcinoma tissue imaging in partial nephrectomy for renal cell carcinoma patients.

## Figures and Tables

**Figure 1 ijms-23-07228-f001:**
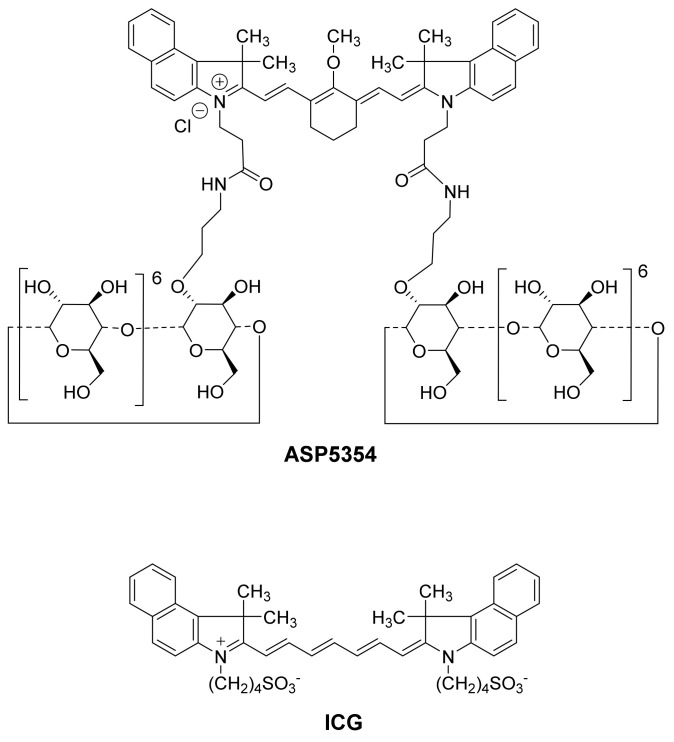
Chemical structures of ASP5354 and indocyanine green (ICG).

**Figure 2 ijms-23-07228-f002:**
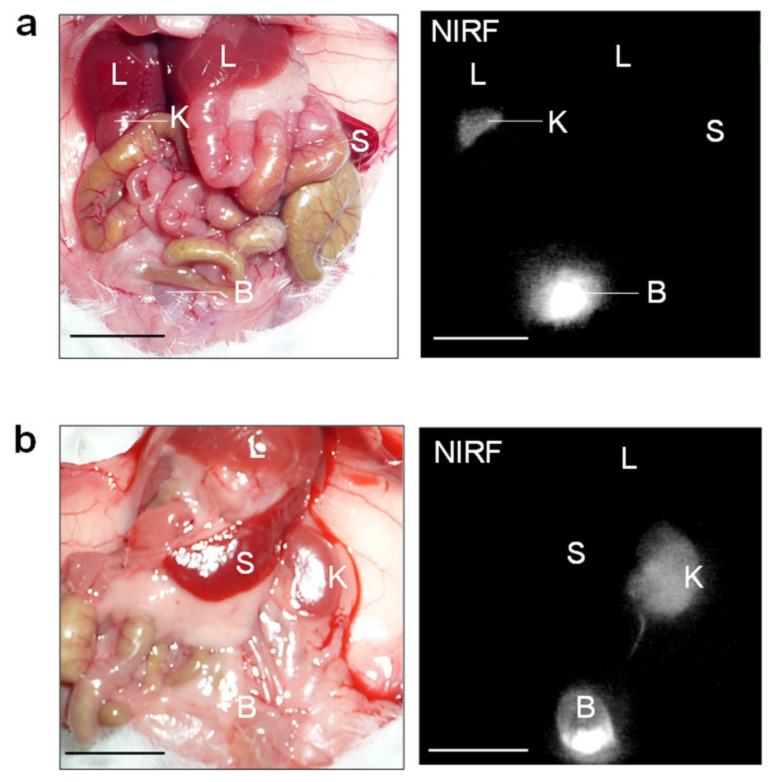
Representative in vivo near-infrared fluorescence (NIRF) imaging (**a**,**b**) of internal organs. Image was obtained at 10 min after the intravenous administration of ASP5354 (12 nmol/kg body weight) in healthy mice. NIRF is displayed in white. Scale bar = 10 mm. B, bladder; K, kidney; L, liver; S, spleen.

**Figure 3 ijms-23-07228-f003:**
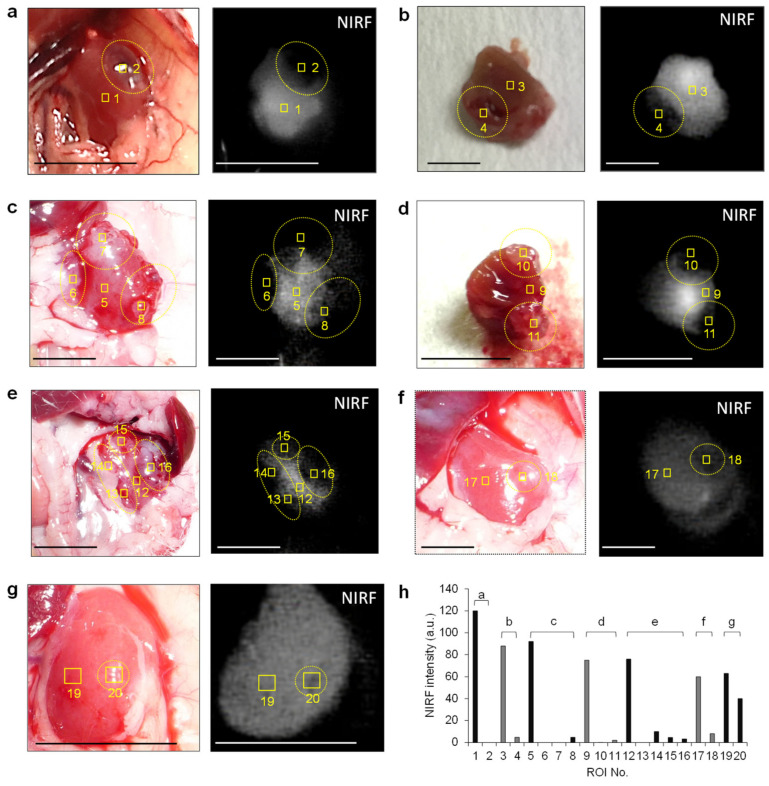
In vivo and ex vivo near-infrared fluorescence (NIRF) imaging of cancerous kidneys. Images were obtained at 10 min after the intravenous administration of ASP5354 (12 nmol/kg body weight) in mice. (**b**) Section of the cancerous kidney shown in image (**a**). (**d**) Section of the cancerous kidney shown in image (**c**). (**h**) Mean NIRF intensity in ROI shown in squares in images (**a**–**g**). Cancer tissues are in the dotted circles. NIRF is displayed in white. Scale bar = 10 mm.

**Figure 4 ijms-23-07228-f004:**
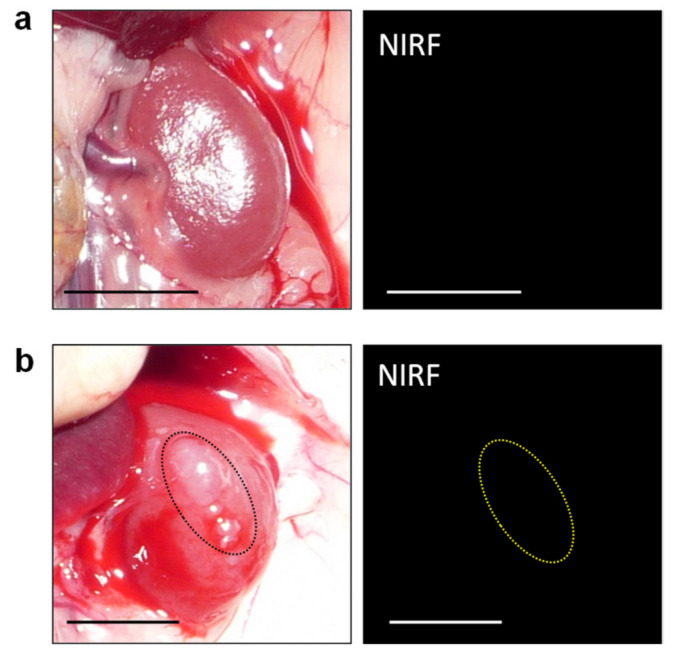
Representative real-time near-infrared fluorescence (NIRF) imaging of healthy (**a**) and cancerous (**b**) kidneys. Images were obtained at 6 h after the intravenous administration of ASP5354 (12 nmol/kg body weight) in mice. Tumor is within the dotted circle. Scale bar = 10 mm.

**Figure 5 ijms-23-07228-f005:**
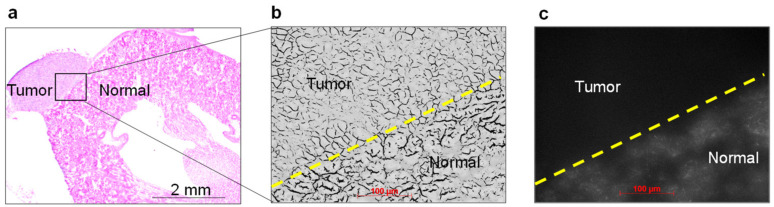
Histopathological analysis of the cancer tissue harvested from cancerous kidney after 10 min post-ASP5354 intravenous administration. (**a**) Hematoxylin and eosin (H&E)–stained tissue. (**b**) Monochrome image of boundary area between tumor and normal tissues in frozen section adjacent to the section used in (**a**). Yellow dashed line is the boundary between the tumor and normal tissues. (**c**) Near-infrared fluorescence image of section (**b**) by ASP5354. NIRF is displayed in white. Yellow dashed line is the boundary between the tumor and normal tissues.

**Figure 6 ijms-23-07228-f006:**
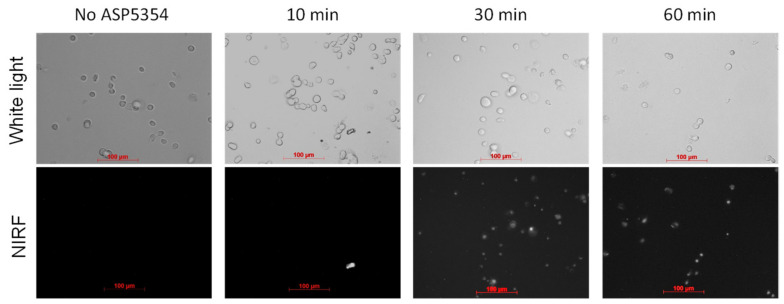
Near-infrared fluorescence (NIRF) imaging of ASP5354 uptake in RenCa cells. NIRF images of cells with 60 s exposure time after incubation with ASP5354 (24 μM) for 10, 30, or 60 min. NIRF is displayed in white.

**Figure 7 ijms-23-07228-f007:**
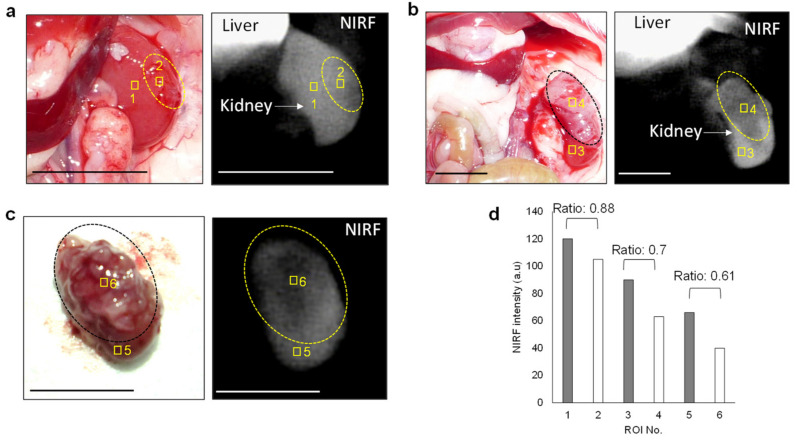
In vivo and ex vivo near-infrared fluorescence (NIRF) imaging of cancerous kidneys using indocyanine green (ICG). Images were obtained at 10 min after the intravenous administration of ICG (120 nmol/kg body weight) in mice. (**d**) Mean NIRF intensity in ROI shown in squares in images (**a**–**c**). Cancer tissues are in the dotted circles. NIRF is displayed in white. Scale bar = 10 mm.

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
