# Peer review of "Near-Infrared Fluorescence Imaging of Renal Cell Carcinoma with ASP5354 in a Mouse Model for Intraoperative Guidance"

_ijms, 2022, doi:10.3390/ijms23137228_

Round 1

Reviewer 1 Report

The paper "Near-infrared fluorescence imaging..." by K. Teranishi deals out the capability of the near-infrared fluorescent iamging agent ASP5354 for in vivo fluorescence imaging of renal cancer. This is crucial problem since clear differentiation between normal anc cancerous tissues could help to identify the localization of diseased porzion of renal tissues. The results, clearly presented, show that ASP5354 fives the possibility to visualize the renal cancer tissues in mouse model immediately after a single intravenous dose, by the near-infrared fluorescence camera system adopted. The author describes clear differences in near-infrared fluorescence intensity between normal and cancer tissues making possible to identify boundaries between normal and cancer tissues in micro and macro near infrared fluorescence imaging demonstrating that ASP5354 is an effective and promising imaging agent for renal cancer.

The paper is well written and it could meet the interest of  a wide range of scientistis involved in teh development of new diagnostic techniques in oncology. I have some doubts about the oragnization of the paper. I suggest the author to anticipale some materials and methods sections before Results section, in order to clarify the same results, understanding of which requires you to read the materials and methods used. After such reorganization, I suggest  to puclish theexcellent  paper.

Author Response

Materials and Methods section was moved before Results section.

Reviewer 2 Report

This study was reported the effectiveness of ASP5354 to distinguish malignant cells from normal tissue in patients with renal cell carcinoma. Overall, this paper is well written. The reviewer thinks that this report has useful information for readers. The reviewer would like to suggest some critiques as follows.

1.     “Renal cancer” and “Renal cancer masses” seem to be inadequate. “Renal cell carcinoma” and “Malignant renal tumors” are better.

2.     On line 72, renal cancer imaging is unclear. The authors should describe the definition of renal cancer imaging.

Author Response

Response 1: In the text and GA, “Renal cancer” and “Renal cancer masses” were changed to “Renal cell carcinoma” and “Malignant renal tumors”, respectively.

Response 2: On line 72, “This study aimed to investigate the usefulness of ASP5354 as an agent for renal cancer imaging in a mouse model.” was changed to “This study aimed to investigate the usefulness of ASP5354 as an imaging agent for NIRF differentiation between renal cell carcinoma and normal tissues in a mouse model.”, in order to definite “renal cancer imaging”.

Reviewer 3 Report

The manuscript of Teranishi is aimed at investigating the suitability of the infra-red fluorescent ASP5354 for in vivoimaging renal cancer. To achieve this goal, the Author applied infra-red imaging of the abdominal cavity in a mouse model of renal cancer as well as infra-red camera analysis of harvested kidney tissues.  The Author concludes that ASP5354 might be potentially be applied in kidney cancer operations in a clinical setting.

The topic fits the scope of the “International Journal of Molecular Sciences”, though it might better suit “Cancers” or “Biosensors”. The manuscript present data in 7 multipanel figures that are prepared in a rigorous fashion and clearly demonstrate the main findings of the manuscript.

The manuscript is written in a clear and conscious manner, citing the relevant references and providing an honest appreciation of the clinical applications of the study results.

This reviewer has no further comments.